# Dynamics of *Podospora anserina* Genome Evolution in a Long-Term Experiment

**DOI:** 10.3390/ijms241512009

**Published:** 2023-07-27

**Authors:** Olga A. Kudryavtseva, Evgeny S. Gerasimov, Elena S. Glagoleva, Anna A. Gasparyan, Saveliy M. Agroskin, Mikhail A. Belozersky, Yakov E. Dunaevsky

**Affiliations:** 1Faculty of Biology, Lomonosov Moscow State University, 119991 Moscow, Russia; for-ol-ga@yandex.ru (O.A.K.); jalgard@gmail.com (E.S.G.); yuga@mail.ru (E.S.G.); g.anna07@mail.ru (A.A.G.); a.sava040102@gmail.com (S.M.A.); 2A.N. Belozersky Institute of Physico-Chemical Biology, Lomonosov Moscow State University, 119991 Moscow, Russia; mbeloz@belozersky.msu.ru

**Keywords:** experimental evolution, long-term experiment, parallel evolution, *Podospora anserina*

## Abstract

The *Podospora anserina* long-term evolution experiment (PaLTEE) is the only running filamentous fungus study, which is still going on. The aim of our work is to trace the evolutionary dynamics of the accumulation of mutations in the genomes of eight haploid populations of *P. anserina.* The results of the genome-wide analysis of all of the lineages, performed 8 years after the start of the PaLTEE, are presented. Data analysis detected 312 single nucleotide polymorphisms (SNPs) and 39 short insertion-deletion mutations (indels) in total. There was a clear trend towards a linear increase in the number of SNPs depending on the experiment duration. Among 312 SNPs, 153 were fixed in the coding regions of *P. anserina* genome. Relatively few synonymous mutations were found, exactly 38; 42 were classified as nonsense mutations; 72 were assigned to missense mutations. In addition, 21 out of 39 indels identified were also localized in coding regions. Here, we also report the detection of parallel evolution at the paralog level in the *P. anserina* model system. Parallelism in evolution at the level of protein functions also occurs. The latter is especially true for various transcription factors, which may indicate selection leading to optimization of the wide range of cellular processes under experimental conditions.

## 1. Introduction

Over the past 20 years, laboratory experimental evolution has rapidly developed [1]. The impetus for the flourishing of this particular scientific direction was the widespread introduction of high-throughput sequencing technologies (next-generation sequencing, NGS), which reads the entire genomes of any living organisms in a quick and cheap manner. Newly emerged mutations can be detected in the genomes of evolving lines by comparing them with the original ancestral population [2]. Consequently, an unprecedented amount of empirical evolutionary information is available for analysis today [3].

A description of the evolutionary dynamics of living systems, as well as an understanding of how predictable evolution can be, serves as the basis for the further development of various scientific areas important in people’s daily lives. Among other things, this includes preparation for the emergence of new pathogens, for the development of antibiotic resistance in pathogens, and for the selective destruction of therapy-evading cancer cells [4]. To study evolutionary processes in depth, various laboratories conduct massively parallel evolution experiments [3], primarily in microbial populations [4]. This direction has received the informal name “evolution-in-a-flask”. Different authors have used viruses, bacteria, unicellular algae, yeast, and filamentous fungi as objects of study [5,6]. The short generation time of microorganisms allows up to dozens of generations of evolution to take place every day. Microbial evolution experiments often aim to contribute to the development of evolutionary theory or to trace changes in individual traits. At the same time, experimental evolution also serves as a tool that can be used to purposefully create organisms for specific biotechnological applications [7].

The vast majority of evolutionary experiments are short-term, typically only lasting a few weeks or months [1]. If applied work is aimed at quickly obtaining the desired phenotype and does not imply long-term adaptive selection, then fundamental research, on the contrary, can potentially be unlimited in time. Long-term experiments are still few in number, as they are extremely laborious. However, such works are highly important for describing evolutionary dynamics and establishing the corresponding molecular mechanisms. Empirical data show that some phenomena only manifest themselves over long time distances [8]. Theoretically, the duration of an evolutionary experiment is only limited by the ability of the experimenter to regularly transfer the studied lines to a fresh nutrient medium [7]. Today, it is an established fact that adaptive evolution is a dynamic long-term process even in the simplest experiments, designed with a single clone (or genotype) as the starting “population”. Such conclusions were made from evolutionary model species. *Escherichia coli* and yeast are the most commonly used among them [9]. However, many interesting results have been obtained through the use of various, less popular organisms. Despite the fact that long-term evolutionary experiments are relatively few, they are very diverse both in terms of objects of study and adaptation conditions. If we wish to place evolution in a predictive context, we need to take a closer look at significantly long-term multiyear evolutionary experiments, which are conducted under controlled laboratory conditions.

We are currently conducting the *Podospora anserina* long-term evolution experiment (PaLTEE)—the only long-term experiment on a filamentous fungus. *P. anserina* is a model ascomycete fungus that demonstrates pronounced mycelia senescence when grown on solid media, but has an unlimited lifespan in submerged culture. To study the genetic aspects of *P. anserina* adaptation to a liquid medium, the long-term evolutionary experiment on eight independent lines was initiated in 2012. During the first 4 years of the experiment, at selected time points, we tested whether de novo mutations were fixed in our model system. Using two different sequencing methods, we demonstrated the fixation of single-nucleotide polymorphisms (SNPs) and short insertion and deletion mutations (indels) in the genomes of all of the experimental populations [10]. Thus, the *P. anserina* model system was assessed as promising, and we decided to continue the experiment. It should be noted that the primary adaptation of *P. anserina* to experimental growth conditions was accompanied by a significant increase in biomass accumulation in the passage. In addition, after the first few passages, rapid morphological changes occurred in all experimental mycelial populations, which lead to the formation of a certain phenotype. Such changes were described with great care in the course of our preliminary short-term *P. anserina* experiment [11]. By starting the PaLTEE, we noted similar phenotypic changes in all independent lineages [10]. Thus, the visible changes were reproducible not only in parallel cultivated lines, but also in independent experiments on the adaptation of *P. anserina* to submerged cultivation.

The purpose of this work is to trace the dynamics of the accumulation of mutations in the genomes of experimental populations of *P. anserina* over a long time interval. To achieve this goal, we continued the evolutionary experiment for another 4 years under the same conditions. During this time, genome-wide sequencing of all fungal lines was carried out twice. The data obtained are presented and analyzed in this publication.

## 2. Results

### 2.1. General Overview of the P. anserina Long-Term Evolution Experiment

The PaLTEE was started in 2012. Two *P. anserina* wild-type strains (founder genotypes), each of which originated from a single mononuclear ascospore, gave rise to eight experimental populations. The genotype denoted as A gave rise to five and the genotype denoted as B gave rise to three experimental populations. We denoted these populations as ‘lineages A1–A5’ and ‘B1–B3’, respectively. Continuous mycelia growth was maintained on the standard liquid medium by synchronous serial passages from flask to flask once every 5 or 6 days. After 4 years of the experiment, the full genomes of all of the lineages without exception were read using high-throughput sequencing (mycelia samples were taken from passage no. 268). As a result, SNPs and indels were detected [10]. Over the next 4 years, the whole genome sequence of all of the lineages was performed two more times (for passages no. 400 and no. 532). Thus, the distance between the analyzed time points was 132 passages. 

The presence of numerous additional potential substitutions in the evolved genomes made us reconsider the methodological approach toward the use of the more accurate filter for the selection of true de novo mutations. Thus, in this report, we present the results of genome-wide analysis (deep WGS) obtained in the course of the first 8 years of the evolutionary experiment with the fungus *P. anserina*. We compare three time points corresponding to passage no. 268 (~4 years of cultivation), no. 400 (~6 years of cultivation), and no. 532 (~8 years of cultivation), using a fairly strict mutation selection approach designed by us specifically for this purpose. To filter out mutations that could be present in founder genotypes at low or very low frequency, we performed a rigorous selection of SNVs (both SNPs and short indels) with the procedure described below. 

### 2.2. Obtaining a Reliable Set of Allele Fixations 

We obtained a SNPs/indels set, which did not contain any polymorphisms, which are not de novo mutations, using the filtering method technically described in the Methods section. We considered only variants that were located in genomic positions visible in all three sequenced time points in the examined lineage. If any sequenced sample from the currently examined lineage in the genomic locus had insufficient read coverage, this genomic position was excluded from analysis, as it was impossible to reliably determine the absence of a low-frequency variant in this sample. Such a procedure guarantees that all variants in the set are true de novo mutations, which means that they do not present at a low frequency in any sequenced sample. Similarly, we filtered indels; however, for each indel, the genomic region of +10/−10 nucleotides around the potential indel was examined for coverage and for the absence of any other low-frequency variants.

In total, we found 312 de novo SNP and 39 de novo indel fixation events, which means that mutations continue to occur in all experimental lineages, as expected. There was a clear trend toward a linear increase in the number of SNPs depending on the number of passages made. We assessed the rate at which mutations were fixed in the lineages (Figure 1A) and found that the overall rate was linear for all experimental lineages, except B2, in which mutations occurred at a much higher rate. This type of dynamics may indicate the fixation of the mutator allele. Interestingly, the mutation rate for indels seemed not to be linear for all lineages; B2 did not look much different from the other lineages.

The relative mutation rates of all nucleotides for observed fixations are shown in Figure 1B, indicating that the C→T/G→A transition is the most common type of nucleotide change (41%). A similar pattern was previously observed in mutation accumulation in lines in an experiment on yeasts [12].

We carefully examined the distribution of de novo mutations fixed during cultivation time across the genome. The diagram in Figure 1C shows that variants are distributed over chromosomes with equal density. No particular places in the genome are found to accumulate mutations faster than others; unfiltered SNPs are also distributed almost equally along the genome.

### 2.3. Characterization of De Novo Mutations Accumulated in Experimental Lineages

Among the 312 SNPs, 153 were fixed in the coding regions of the *P. anserina* genome. Considering the fact that the coding sequences together make up around 47% of the genome examined, our data indicate an almost equal probability of fixing point substitutions in coding and non-coding regions. Only 38 synonymous mutations (also called first approximation “silent” mutations) were found among 153 gene substitutions. The evident predominance of significant substitutions suggests that at least some of them may be adaptive under experimental conditions. In other words, they did not arise by chance. Among them, 42 were classified as nonsense mutations (resulting in a stop codon) and 73 as missense mutations (resulting in an amino acid non-synonymous change in the corresponding protein). Slightly more than half of the indels identified (21 out of 39) were also localized in coding regions. Most of them (17 out of 21) were frameshift mutations (leading to a shift in the reading frame), which means a probable loss of function of the protein product. The simple empirical rule may be established: once fixed, the mutation persists over time. It works for almost all mutations observed in the PaLTEE.

### 2.4. Parallel Evolution

Parallelisms have been identified in fixing mutations in the same genes of independent *P. anserina* populations. Eight protein-coding sequences evolved in two or more lines. With our substitution filtering method, the most outstanding result of fixing new alleles was six out of eight experimental lines. Moreover, such a high substitution number was achieved in two different genes. One of them is responsible for coding putative guanine nucleotide-binding protein alpha-1 subunit (gene Pa_7_7970); the other encodes a putative protein with an unknown function (Table 1). Of note, the parallel evolution of Pa_7_7970 was described in our fourth-year report [10].

Furthermore, we report the parallel evolution in another guanine nucleotide-binding protein Pa_1_23950, which might be considered a paralog of Pa_7_7970. Here, we confirm these results and also add a few new cases of the parallel evolution of paralogs. This category includes sixteen putative protein-coding *P. anserina* genes, which form five separate groups of homologs. What is more, the homologs within each group are fairly close to each other (Table 2). According to bioinformatics prediction, the putative protein products of the paralogs described perform exactly the same or very similar biological functions.

## 3. Discussion

The current work presents a sequel for our ongoing long-term evolutionary experiment, where we constantly monitor genomic changes during the adaptation of a filamentous fungus to new environmental conditions. With three experimental time points examined using NGS deep sequencing, the linear trend of mutation fixation became obvious. The striking exception was experimental lineage B2, which demonstrated a superlinear fixation rate. Interestingly, the raw number of substitutions detected by the SNP caller before filtering was applied and was not greater for samples from lineage B2 and depended more on the average genome coverage level. 

The notable feature of both fixed substitutions and indels is their random distribution across the genome: the proportion of mutations correlates well with chromosome length (Figure 1C), and it also seems equally likely to obtain a fixed mutation in coding and non-coding regions. In contrast, our analysis of mutation effects revealed that most of them are non-synonymous substitutions with notable cases of parallel evolution, which clearly points to positive selection. We can hypothesize that mutations in non-coding regions might also be functional, affecting control elements such as promotors or enhancers.

The most suitable for comparative analysis is the work presented in [4], performed on *Saccharomyces cerevisiae*. This is the longest evolutionary experiment ever carried out on budding yeast. As in our model system, experimental populations fixed mutations throughout the experiment. At the same time, in contrast to *P. anserina*, among 90 focal experimental *S. cerevisiae* populations, no population with highly elevated mutation rates appeared. Attention is drawn to the fact that, in both model systems, the percentage of missense mutations had very similar meanings. It was 45–50% across different *S. cerevisiae* strains and the experimental environments used, with 42% in *P. anserina*. Conversely, the proportion of other groups of mutations significantly differed. The total contribution of nonsense and indel mutations in *P. anserina* was about five times higher than in *S. cerevisiae*. Most likely, the filamentous fungus is subjected to a more pronounced selection than yeast.

The transition to transversion ratio for observed SNPs is equal to 1.1, which is close to the *S. cerevisiae* value determined in a mutation accumulation experiment described by Zhu et al. [12]. The ratio does not change during cultivation time. The commonly observed value of 2.1 is typical of coding regions of the genome, where transversions most likely lead to amino acid changes, which in turn most likely lead to a dangerous loss of function, therefore making transition mutations less likely to be fixed in the population. In the case of uniform rates of both mutation and fixation (ignoring genome composition bias), the expected transition to transversion ratio should be 0.5. The value obtained in our experiment can be interpreted as a strong shift toward the fixation of non-synonymous mutation types. This is in good agreement with the low proportion of silent mutations among all fixations, which we described above. 

The parallel evolution of genes was initially described in our first report [10], with very notable cases of the parallel evolution of guanine nucleotide-binding proteins, which can be considered as a parallel evolution of paralogs. Currently, we can only add putative polyphosphoinositide phosphatase Pa_6_2900 to this list, due to the fact that a second mutation was detected at passage no. 400, while the first was already visible at passage no. 268. This means that mutations in these genes rapidly occurred and most of them were fixed in the first four years of the experiment. This might be due to the high active positive selection pressure acting on this group of genes.

Parallel evolution at the level of protein functions was also found in the *P. anserina* model system. The putative products of a number of genes involved in the same biological processes underwent changes. We emphasize that genes of this type are presumably not homologous to each other. Attention is drawn to a significant proportion of substitutions in putative transcription factors. Almost all of the mutations of transcription factor genes potentially affect their product function. Many of the factors reported contain an annotated DNA-binding domain. Focusing on the regulation of RNA synthesis may indicate that the selection occurring during the PaLTEE is aimed at the optimization of a wide range of cellular processes. Indeed, the *P. anserina* under study is forced to adapt to very unusual conditions. Specifically, the fungus lives in a closed environment within a liquid nutrient medium, which is constantly subjected to intensive aeration. The fungi do not encounter anything like this in nature.

## 4. Materials and Methods

### 4.1. Evolutionary Experiment Setup

The data presented in the current paper were obtained during the long-term evolutionary experiment started in 2012 [10]. Briefly, two homokaryotic vegetatively incompatible wild strains, A and B, gave rise to five and three independently grown experimental populations, denoted as A1–A5 and B1–B3. The initial strains A and B used in this work are sexual descendants of widely used laboratory strains s and S, respectively, which were kindly provided by Annie Sainsard-Chanet and Carole H. Sellem (Département Biologie Cellulaire et Intégrative, Centre de Génétique Moléculaire, CNRS, Gif-sur-Yvette Cedex, France). The submerged cultivation of eight experimental lineages was carried out in 750 mL Erlenmeyer flasks with 100 mL of standard M2 medium [12] on the rotor shaker at 27 ± 2 °C. Synchronous serial passages were carried out every 5–6 days. The whole-genome sequencing of both founder and all the experimental lineages at passage no. 268 is described in [10]; passages no. 400 and no. 532 are presented in this work.

### 4.2. DNA Extraction and Libraries Construction

DNA was extracted using the CTAB-based method [13] with slight modifications. Small pieces of mycelia (approximately 0.5 cm^3^) were ground in liquid nitrogen in a mortar and pestle. An amount of 1 mL of 2× CTAB isolation buffer (100 mM Tris-HCl, pH 8.0, 1.4 M NaCl, 20 mM EDTA, 2% hexadecyltrimethylammonium bromide (CTAB), 1% Na_2_SO_3_, 1% polyvinylpyrrolidone, 0.2% 2-mercaptoethanol) was added to the ground mycelia. Samples were transferred into 2.0 mL Eppendorf tubes and incubated at 56 °C for 60 min. Then, 2/3 volume of chloroform was added and gently but thoroughly mixed. Then, samples were centrifuged for 5 min (13,000× *g*) (MiniSpin, Eppendorf, Hamburg, Germany) to concentrate the layers. The upper aqueous phase was transferred to clean 2.0 mL Eppendorf tubes. The extraction with chloroform was repeated one more time. The upper aqueous phase was transferred to clean 1.5 mL Eppendorf tubes. One volume of cold isopropanol was added, mixed, and incubated (−20 °C) for 20 min to precipitate nucleic acids. After precipitation, samples were centrifuged for 10 min 13,000× *g*. The upper phase was carefully removed. The precipitate was twice washed with 0.5 mL of 80% ethanol (v/v) and spun for 3 min at 13,000× *g* each time. Ethanol was removed for dryness. The precipitates were dried for 5–10 min at room temperature and then dissolved in 50–150 μL nuclease free water. DNA yield was about 40–100 ng/μL. After the ultrasonic fragmentation of isolated genomic DNA with Covaris S220 (treatment time 40 s, 175 peak power), the sequencing library with an insert size of 300–400 bp was prepared using a NEBNext^®^ Ultra™ DNA Library Prep Kit for Illumina (New England Biolabs) according to the manufacturer’s instructions [14]. DNA input in the end prep reaction was about 500–700 ng. The input of adaptor-ligated DNA in the PCR enrichment reaction was about 16–19 ng and the number of PCR cycles was 10. The concentration of the obtained libraries was about 20–50 ng/µL. The quality of the libraries was assessed using the 2100 Bioanalyzer system (Agilent Technologies, Santa Clara, CA, USA) and qPCR.

### 4.3. Whole-Genome Sequencing

Sequencing libraries were sequenced using an Illumina HiSeq 4000 instrument (Illumina, San Diego, CA, USA) in paired-end mode with a read length of 151 bp. For passage no. 400, the sequencing yield was from 5 to 10 million of paired reads (average 6.8 million), and for passage no. 532 from 24 to 31 million paired reads (average 26.7 million). Data are deposited under BioProject PRJNA984431.

### 4.4. Small Variations Fixation Analysis

Sequencing reads for 268, 400, and 532 passages were obtained using the Illumina platform in paired-end sequencing mode, so we used a similar read processing pipeline for SNV calling. Read quality was assessed with FastQC v0.11.9 [15] and quality trimming was carried out using Trimmomatic v.039 [16]. Quality- and adaptor-trimmed read pairs were mapped on the reference genome of the *P. anserina* strain S mat+ [17], downloaded from NCBI GenBank (assembly accession GCA_000226545.1) with bowtie2 v.2.4.1 [18], and read mapped in ‘--end-to-end’ mode with the ‘--very-sensitive’ option. Read alignments were filtered and pre-processed with Samtools and BCFtools v.1.19 [19,20] and converted into sorted binary alignment files. Optical duplicates were removed with the Picard tool ‘MarkDuplicates’. Variants were called using Freebayes v.0.9.21 [21] with ‘--no-complex --min-coverage 10 --min-alternate-fraction 0.01 --min-base-quality 5 --min-alternate-count 7 --pooled-continuous’. Pileup files for each sample were generated with Samtools mpileup.

An initial set of candidate SNPs was obtained by gathering all SNPs with an alternative allele frequency of >85% in any sample. This set was subjected to a custom filtration pipeline. A brief description of the filtering rules is given below. Coverage in each sequenced sample, including the founders (strains A and B), is calculated with the Samtools ‘depth’ program with the ‘-aa’ option to write down coverage in each genomic position. For each SNP, from the initial SNPs set, the founder sample and the three respective time points are selected for analysis. For example, if the SNP from the set is found in time point 400 of lineage A1, then founder A time points 268, 400, and 533 for A1 are selected. The rules to discard the SNP are the following: (i) coverage in at least one selected sample is less than 10 reads; (ii) the SNP is observed in the founder sample; (iii) the alternative allele is observed at low frequency (<85%) in at least one of the selected samples. We randomly took 40 filtered SNPs and performed manual curation of read alignments near the variant position in a series of samples in IGV [22]. 

A similar algorithm was used to determine short indel fixation events. However, for the indel filtering, we examined the region (−10; +10) base pairs around the detected indel coordinate. All indels in adjacent positions were counted on the filtration step to remove possible calling artifacts; the indel was falsely broken in a combination of SNP + a shorter indel in some of the samples. 

The final results were gathered in tables (Appendix A) with a custom Python script. To construct the figure, we used Python, where we applied regular graphics “matplotlib”.

### 4.5. Functional Annotation and Annotation of Possible Effects

The effect of SNP or indel observed was determined with a custom Python script, which inserted the variation into the annotated gene’s sequence and performed the in silico translation of the modified mRNA. The results for SNPs were further re-checked with the snpEff v.5.1 [23] tool. 

The possible gene function was assigned by finding the best protein alignments in UniProtDB with blastp from ncbi-blast-suite suite v.2.5.0+ [24] with the options ‘-evalue 1 × 10^−6^ -max_target_seqs 1’. Protein domains were determined with the hmmer v.3.3.2 [25] package using the Pfam-A database. The results of the functional annotation were collected with custom scripts.

## Figures and Tables

**Figure 1 ijms-24-12009-f001:**
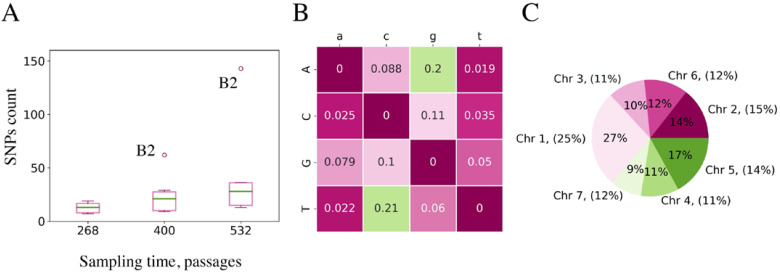
(**A**) Growth of the number of fixed SNPs observed in sampled time points. Boxplot represents the distribution for all lineages with the exception of B2 in 400 and 532 time points. Lineage B2 is an outlier with a much higher number of fixed SNPs. In 400 and 532 points, the number of mutations in B2 is shown with a circle. (**B**) Substitution matrix for fixed SNPs observed in all samples. The frequency of substitution is shown for each event, where the nucleotide in a column (lowercase letter) mutates into the nucleotide in a row (uppercase letter). (**C**) Fixed SNPs distribution per chromosome. The percentage of SNPs from the total number of observed SNPs is shown inside the pie chart for each chromosome. Outside the pie chart, the chromosome name and the percentage of the total genome length occupied by the chromosome are given.

**Table 1 ijms-24-12009-t001:** Parallelisms in alleles in independent *P. anserina* lines.

Gene ID	Putative Protein Function	The Number and Type of Independent Fixations
PODANS_3_400	Uncharacterized protein	6 frameshifts
PODANS_7_7970	Guanine nucleotide-binding protein alpha-1 subunit	6 missense
PODANS_1_23950	Guanine nucleotide-binding protein alpha-3 subunit	3 silent
PODANS_6_2900	Polyphosphoinositide phosphatase	2 missense
PODANS_1_700	Uncharacterized regulator of G protein signaling	1 nonsense, 1 in-frame
PODANS_2_890	Cell pattern formation-associated protein stuA OS	1 missense, 1 frameshift
PODANS_1_10140	Transcriptional regulatory protein pro-1 (fungal specific transcription factor)	1 nonsense, 1 frameshift
PODANS_2_6940	Small domain found in the jumonji family of transcription factors	1 in-frame, 1 frameshift

**Table 2 ijms-24-12009-t002:** Parallelisms in *P. anserina* paralogs.

Groups of Homologs, Gene ID	The Number and Type of Independent Fixations	Putative Protein Function
PODANS_7_7970 PODANS_2_10260 PODANS_1_23950	6 missense 1 missense 3 silent	Guanine nucleotide-binding protein alpha-1 subunit Guanine nucleotide-binding protein alpha-2 subunit Guanine nucleotide-binding protein alpha-3 subunit
PODANS_3_2010 PODANS_5_1930	1 missense 1 nonsense	Transmembrane transporter activity Transmembrane transporter activity
PODANS_5_3180 PODANS_2_6680	1 nonsense 1 silent	Vegetative incompatibility protein HET-E-1 Vegetative incompatibility protein HET-E-1
PODANS_4_40 PODANS_5_6830 PODANS_5_3750 PODANS_5_7750 PODANS_6_10330 PODANS_5_6830	1 missense 1 missense 1 missense 1 silent 1 silent 1 missense	Putative polyketide synthase fatty acid biosynthetic process (conidial yellow pigment) Putative dual specificity polyketide synthase secondary metabolite biosynthetic process (lovastatin nonaketide synthase) Putative polyketide synthase (lovastatin nonaketide synthase) Putative polyketide synthase (lovastatin nonaketide synthase) Putative polyketide synthase (lovastatin nonaketide synthase) Putative dual specificity polyketide synthase secondary metabolite (lovastatin nonaketide synthase)
PODANS_4_6070 PODANS_1_15320 PODANS_3_3480	1 nonsense 1 missense 1 nonsense	Cyclin-dependent kinase 1 Protein kinase wis1 Serine/threonine-protein kinase

## Data Availability

Not applicable.

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
