# Peer review of "Dynamics of Podospora anserina Genome Evolution in a Long-Term Experiment"

_ijms, 2023, doi:10.3390/ijms241512009_

Round 1

Reviewer 1 Report

The authors declared, “Dynamics of Podospora anserina genome evolution in the long-term experiment”. Despite the importance of the study, It has many grammar and language mistakes.

The order of event writing should be the same either in the abstract, introduction, material, ……….so on.

An expert in the English language should revise it before publishing. The following minor points must be taken into consideration:

Abstract:

-The abstract should briefly state the purpose of the research (illustrating the aim of the work).

- line 11 haploid independent changed to haploid- independent

- key words need to be modified

-Arrange the keywords in alphabetic order

Introduction:

-The introduction needs to be more informative.

Results

-  The results are very well written, and the figures are excellent but please provide the program used in these figure in the material section

- Line 107  reliable  changed to reliably

- Line towards changed to toward

Discussion and conclusion

-Authors need to improve discussion and explore the significance of the results of the study compared to other studies.

Extensive editing of English language required

Author Response

We are grateful to the reviewer for the efforts made to improve the manuscript.
We have tried to fix all the errors and answer the questions raised.

The order of event writing should be the same either in the abstract, introduction, material, ……….so on.

We tried to adhere to this order  

An expert in the English language should revise it before publishing. The following minor points must be taken into consideration:

We consulted and tried to improve the quality of the English language as much as possible

-The abstract should briefly state the purpose of the research (illustrating the aim of the work).

The purpose of the work аdded 

- line 11 haploid independent changed to haploid- independent

Fixed 

- key words need to be modified

Keywords modified  

-Arrange the keywords in alphabetic order

The keywords are arranged in alphabetical order 

-The introduction needs to be more informative.

We have added additional information  

-  The results are very well written, and the figures are excellent but please provide the program used in these figure in the material section

Data added in the materials section  

- Line 107  reliable  changed to reliably

Fixed  

- Line towards changed to toward

Fixed  

-Authors need to improve discussion and explore the significance of the results of the study compared to other studies.

We tried to improve the discussion by adding additional material

Extensive editing of English language required

We consulted and tried to improve the quality of the English language as much as possible

Reviewer 2 Report

the study reports on a long-term study of genome evolution of the fungus Podospora anserina, a relevant model fungus. The main question addressed is which mutations appear and get fixed in the genome over many generations under controlled growth conditions.

The manuscript clearly addresses a gap of kowledge in the field as very few is known about long-term mechansims operating in the evolution of fungi (as compared to a bulk of knowledge avaialble concerning short term appearance of mutations leading to specific singular effects as, e.g., increased fungicide ressistance).

The most interesting outcome reported is that in the long-term perspective there appears to be a clear tendency towards equal rates of mutation fixation in parts of the genome considered coding and non-coding. This is not in line with expectations derived from short-term observations.

The methodology of the study is absolutely adequate and results and arguments are clearly presented. References are appropriate, too.

Supplementary and non-published Tables might be organized or re-sized in a way to avoid disruption by page borders (landscape view, smaller character size).

Very interesting update from long-term Podospora genome evolution experiment. Congratulations, please go ahead.

Some typos, for instance lines 175/176 should probably read "..., where we constantly ..." ?

Please check the manu once more carefully.

Author Response

We are grateful to the reviewer for the efforts made to improve the manuscript. We have tried to fix all the errors and answer the questions raised.

Supplementary and non-published Tables might be organized or re-sized in a way to avoid disruption by page borders (landscape view, smaller character size).

We have presented additional tables in such a way as to demonstrate all relevant data in a row. We think that perhaps the problem is not related to the source tables, but to the stage of automatic processing of PDF files. Therefore, we can repeat the loading of these tables if we have the ability to load easily readable xls files.

Some typos, for instance lines 175/176 should probably read "..., where we constantly ..." ?

Fixed  

Please check the manu once more carefully. 

Reviewer 3 Report

The manuscript reports on the genome sequencing of strains of the fungus Podospora anserina after a long term evolution experiment.  The experiment has been ongoing for eight years, with here the genomes of eight passaged lineages sequenced.  The analysis of the polymorphisms identified 312 SNPs an 39 indels.  The impacted genes are reported.  Given that most long term evolution studies focus on species that grow as yeast or bacterial forms, this work has a degree of novelty that would interest those doing similar research especially as the same gene (or pathways) are mutated in independent lineages.

However, the major issue of the research is that there is no phenotype reported for the strains!  What is therefore being evolved here, or is this actually the process of random mutation and fixation in action?

There are a number of minor edits worth considering.

Line 4: remove space.

Line 22: spelling ‘anserina’.

Line 30: ‘organism’.

Line 95: ‘of the first 8 years’.

Line 102: likely needs some extra text or words rather than ‘We obtained highly confident SNPs/indel set’.

Line 109: ‘in the set’.

Line 110: ‘for each indel’.

Line 111: ‘was’ for ‘were’.

Line 119: ‘mutator allele’; this comment/suggestion can easily be checked by looking at potential genes involved in this process.  Is there an obvious reason B2 looks different from the other strains?

Line 140: ‘along the genome’.

Line 148: ‘they did not arise by chance’ is unlikely correct – don’t they arise by chance and then are subject to positive selection?

Line 165: delete space.

Line 167: ‘considered a paralog’.

Line 191: ‘the Saccharomyces’ and ‘in a mutation’.

Line 223: ‘in the current paper were obtained’.

Line 237: ‘3’ superscript.

Line 239: ‘SO3’

Line 245: delete ‘c’.

Line 249: ‘spinned’.

Line 264: ‘Data are deposited’.

Line 279: ‘An initial set of candidate SNPs was obtained’.

Line 280: ‘to a custom’.

Line 289: ‘took’ for ‘taken’.

Figure 1 is likely too small to feature the details in a printed version of the paper.

Author Response

We are grateful to the reviewer for the efforts made to improve the manuscript. We have tried to fix all the errors and answer the questions raised.

However, the major issue of the research is that there is no phenotype reported for the strains!  What is therefore being evolved here, or is this actually the process of random mutation and fixation in action?

This issue was studied in great detail by us earlier - in our short-term preliminary experiment on the same model fungus.  A detailed description of morphological changes can be found in [11]. In running this long-term experiment, we noted that from a phenotypic point of view, all experimental lines reproduced the same changes, albeit at different rates. This fact is described in the article [10]. We have added additional information in the introduction.

Line 4: remove space.

Fixed  

Line 22: spelling ‘anserina’.

Fixed  

Line 30: ‘organism’.

Fixed

Line 95: ‘of the first 8 years’.

Fixed  

Line 102: likely needs some extra text or words rather than ‘We obtained highly confident SNPs/indel set’.

Рhrase corrected    

Line 109: ‘in the set’.

Fixed  

Line 110: ‘for each indel’.

Fixed  

Line 111: ‘was’ for ‘were’.

Fixed  

Line 119: ‘mutator allele’; this comment/suggestion can easily be checked by looking at potential genes involved in this process.  Is there an obvious reason B2 looks different from the other strains?

If we turn to the phenotypic characteristics, the B2 lineage also does not have any obvious morphological differences from the other experimental populations.

Line 140: ‘along the genome’.

Fixed  

Line 148: ‘they did not arise by chance’ is unlikely correct – don’t they arise by chance and then are subject to positive selection?

Вad phrase removed 

Line 165: delete space.

Fixed  

Line 167: ‘considered a paralog’.

Fixed  

Line 191: ‘the Saccharomyces’ and ‘in a mutation’.

Fixed  

Line 223: ‘in the current paper were obtained’.

Fixed  

Line 237: ‘3’ superscript.

Fixed  

Line 239: ‘SO3’

That's right here 

Line 245: delete ‘c’.

Тhe word is finished  

Line 249: ‘spinned’.

Fixed  

Line 264: ‘Data are deposited’.

Fixed  

Line 279: ‘An initial set of candidate SNPs was obtained’.

Fixed  

Line 280: ‘to a custom’.

Fixed  

Line 289: ‘took’ for ‘taken’.

Fixed  

Figure 1 is likely too small to feature the details in a printed version of the paper.

Figure enlarged  

Round 2

Reviewer 3 Report

The authors have addressed the point about the missing phenotype data.